# Reaching Frail Elderly Patients to Optimize Diagnosis and Management of Atrial Fibrillation (REAFEL): A Feasibility Study of a Cross-Sectoral Shared-Care Model

**DOI:** 10.3390/ijerph19127383

**Published:** 2022-06-16

**Authors:** Caroline Thorup Ladegaard, Carsten Bamberg, Mathias Aalling, Dorthea Marie Jensen, Nina Kamstrup-Larsen, Christoffer Valdorff Madsen, Sadaf Kamil, Henrik Gudbergsen, Thomas Saxild, Michaela Louise Schiøtz, Julie Grew, Luana Sandoval Castillo, Anne Frølich, Helena Domínguez

**Affiliations:** 1Cardiology Department Y Bispebjerg and Frederiksberg, Hospital Copenhagen Denmark, 2000 Frederiksberg, Denmark; carolineladegaard@hotmail.com (C.T.L.); carsten.bamberg.01@regionh.dk (C.B.); dortheamj@gmail.com (D.M.J.); christoffer.valdorff.madsen.01@regionh.dk (C.V.M.); sadafkamil88@gmail.com (S.K.); 2Department of Biomedicine, University of Copenhagen, 1165 Copenhagen, Denmark; 3VihTek Research Center for Welfare Technology Region, 2600 Hovedstaden, Denmark; mathias.aalling@regionh.dk; 4Innovation and Research Centre for Multimorbidity, Slagelse Hospital, Region Zealand, 4180 Sorø, Denmark; nikam@regionsjaelland.dk (N.K.-L.); anfro@sund.ku.dk (A.F.); 5Section of General Practice, Faculty of Health and Medical Sciences, University of Copenhagen, 1165 Copenhagen, Denmark; 6The Parker Institute Bispebjerg and Frederiksberg Hospital Copenhagen Denmark, Section of General Practice, Department of Public Health, University of Copenhagen, 1165 Copenhagen, Denmark; henrik.gudbergsen@sund.ku.dk; 7Grøndahlslægerne Godthåbsvej 239a Vanløse, 2720 Copenhagen, Denmark; saxild@mac.com; 8Center for Clinical Research and Prevention Bispebjerg and Frederiksberg, Hospital Copenhagen Denmark, 2400 Copenhagen, Denmark; michaela.louise.schioetz@regionh.dk (M.L.S.); julie.grew@regionh.dk (J.G.); 9Geriatrics Department, Bispebjerg Hospital, 2400 Copenhagen, Denmark; luana.sandoval.castillo@regionh.dk

**Keywords:** sensor Holter monitoring, atrial fibrillation, health professionals, frail elderly patients, cross-sector collaboration

## Abstract

**Introduction:** Atrial fibrillation (AF) management in primary care often requires a referral to cardiology clinics, which can be strenuous for frail patients. We developed “*cardio-share*” (CS), a new cross-sector collaboration model, to ease this process. General practitioners (GPs) can use a compact Holter monitor (C3 from Cortrium) to receive remote advice from the cardiologist. **Objective:** To test the feasibility and acceptability of the CS model to manage suspected AF in frail elderly patients. **Methods:** We used a mixed methods design, including the preparation of qualitative semistructured interviews of GPs and nurses. **Results:** Between MAR-2019 and FEB-2020, 54 patients were consulted through the CS model, of whom 35 underwent C3 Holter monitoring. The time from referral to a final Holter report was shortened from a mean (SD) of 117 (45) days in usual care to 30 days (13) with the CS model. Furthermore, 90% of the patients did not need to attend visits at the cardiology clinic. The GPs and nurses highlighted the ease of using the C3 monitor. Their perception was that patients were confident in the GPs’ collaboration with cardiologists. **Conclusions**: The CS model using a C3 monitor for AF is both feasible and seems acceptable to GPs. The elapsed time from referral to the Holter report performed for the diagnosis was significantly reduced.

## 1. Introduction

Atrial fibrillation (AF) is estimated to occur with a prevalence in the Danish population of 0.4–1.0% [1]. AF is the most common sustained cardiac dysrhythmia, and the prevalence increases with age [1,2] and causes immense morbidity [3]. Palpitations, dizziness, fatigue, and significant discomfort are some of the nonspecific and indefinable symptoms that AF triggers. However, approximately 30% of all patients with AF have no symptoms [4,5,6]. Patients with asymptomatic AF pose a diagnostic challenge, and especially frail elderly patients, who have a particularly high risk of stroke [7,8]. Anticoagulant medication can prevent possible stroke once the diagnosis is decided [9,10,11,12,13,14,15,16,17,18,19,20]. The decision regarding the use of oral anticoagulation (OAC) is based on a medical assessment of the benefit of preventing stroke and the concomitant risk of bleeding in the patient [14], as it is recommended by the European Society of Cardiology for the better care of AF [21]. In this assessment, the general practitioners (GPs) will often be of crucial importance in making the correct decision, both regarding the assessment and the possible subsequent treatment, as the GP often knows the patient’s risk of bleeding and preferences of treatment. The examination process might be optimized through “The Cardio-share model”. In the Cardio-Share model (Figure 1), the GP uses the existing opportunity for dialogue with cardiologists at the hospital to ensure that there is an indication for further examination. If indicated, the GP attaches a sensor to monitor the subsequent heart-rhythm advice from the cardiologist who is in dialogue with the patient’s GP. The patient only attends the hospital ambulatory if there is a need for specialized examinations or targeted consultations with a cardiologist.

The Cardio-Share model is, thus, a new organizational model, which is based on the use of a compact heart sensor that registers the heart rhythm. The sensor is placed on the front of the patient’s chest, either by the GP or by a nurse. The sensor registers the heart rhythm for a period from one to seven days. While the patient is wearing the C3 device, the patient can set time stamps by clicking on the device if any unusual or worrying symptoms occur. Recordings are analyzed after resuming recording with the C3 Holter. Follow-up on the sensor measurement results takes place at the patient’s GP office. The Cardio-Share model is based on a close collaboration between the patient, the patient’s GP, and the cardiologists at the hospital. The cardiologist prepares a report and treatment plan. It is possible to conduct a video conference with the patient, GP, and cardiologist to discuss the analysis results and treatment. The aim is to explore whether the Cardio-Share model is feasible for and acceptable to GPs to provide patients with faster AF management, with a reduced burden related to multiple visits to the cardiology clinic. 

The specific questions that this feasibility study aims to elucidate are whether the Cardio-Share model facilitates the evaluation of frail patients for heart arrhythmia with remote support by cardiologists, whether the elapsed time from referral to the result of monitoring can be shortened compared to usual practice, and whether the Cardio-Share model has the potential to be well accepted by GPs.

## 2. Method

The project was designed as a clinical pragmatic study and it includes the criteria of the Pragmatic Explanatory Continuum Summary [22]:

The criteria are: (1) Patients who are evaluated in the project are the same as those who are evaluated in the usual way. (2) Inclusion of patients in the study takes place in connection with a regular consultation in the GP’s office or in the orthogeriatric department. (3) The health professionals who handle the patient’s treatment in the project (the GP and cardiologist at the hospital) are the same as usual. (4) The resources used in the project (which include GPs, the hospital’s cardiologist, the nurses in the cardiology clinic who read the heart-rhythm-monitoring recordings) are the same as in a regular outpatient examination. (5) The proposed “Cardio-Share” model is as flexible as the usual outpatient assessment. (6) Patients follow the same clinical tests as in usual care, except that study patients participate in interviews, which are not normally part of the usual outpatient activities. (7) Primary outcome is clinically relevant for the usual patient assessment procedures (i.e., the time from decision on heart-rhythm monitoring until the diagnosis/conclusion). (8) All data collected in the project are analyzed as part of the primary outcome. 

### 2.1. Study Setting, Participants, and Design

The feasibility and acceptability of the “*Cardio-Share*” model was tested by a mixed method design following an exploratory sequential scheme, with the aim of defining a set of rules for a subsequent pragmatic clinical trial. The study was conducted from the 7th of March 2019 to the 20th of February 2020. The target population was frail elderly patients who were 65 years or older and suspected of having AF from one large general practice with 9.000 registered patients. In the same period, a separate cohort from the orthogeriatric department at Bispebjerg–Frederiksberg University Hospital (BFH) used C3 devices and remote cardiologist support in a similar manner. The GPs could refer patients other than the target population.

When geriatric specialists in the hospital or GPs identified frail elderly patients suspected of AF diagnosis or the adequate management of diagnosed AF, they sent a request to the cardiologist at BFH. The Danish national communication standard (MedCom, DIS90) was used for such requests, and for remote-specialist advice and file exchange. MedCom is integrated in the e-health records that are used in general practice and in the hospitals in the Capital Region (EPIC Systems Corporation, Verona, WI, USA). 

The patients were offered continuous heart-rhythm recording from one to seven days using C3, the compact Holter monitor from Cortrium (Glostrup, Denmark), which could be initiated in the GP office or at their homes. The control group included consecutive patients suspected of AF and referred for standard care Holter monitoring in the outpatient cardiology clinic in the same period.

The outcome measure was the time from when the patient was referred for Holter monitoring to the final Holter report that was used to decide or disprove a heart-rhythm disorder in the intervention group and in the control group.

To assess the acceptability of the “*Cardio-Share*” model, semistructured interviews were conducted with the GPs and nurses in the GP office to obtain their experiences with the “*cardio-share*” model. Interviews were recorded and subsequently transcribed verbatim for analysis.

#### 2.1.1. Inclusion Criteria

Patients were eligible if they were suspected of AF and were ≥65 years of age, and they were considered to be “frail” on the basis of at least one of the characteristics described in Table 1.

Although the frail elderly was the group of interest, inclusion was liberal to explore patient groups that the GP wanted to evaluate for heart-rhythm arrhythmias without referring the patient to the hospital ambulatory, and on patients with known AF that required Holter monitoring to evaluate the AF burden, adequate rate control, or chronotropy questions. Evaluation of cardiac source of stroke elicits more than Holter monitoring and was outside the scope of the study. 

#### 2.1.2. Exclusion Criteria

Patients who did not give signed consent to participate in the project and where follow-up was not possible.

#### 2.1.3. Quantitative and Qualitative Data Stream

The date when each query was sent from the GPs was recorded as the initial date. Subsequently, we recorded the date for the response from the cardiologist, the date of initiation, and the end of the C3 Holter recording, and the date for sending the report and cardiologist advice to the GP. Likewise, we recorded whether there was a need for dialogue before qualifying the indication for Holter monitoring. We also recorded whether there was a reason other than suspicion of atrial fibrillation that elicited the initial query. The patient age, date, known diagnoses, and selected medications were also recorded. Finally, we recorded whether the patient was subsequently referred to the cardiology ambulatory on a regular basis.

The alignment and adjustments of the pathway depicted in Figure 1 to the GP’s workflow were obtained at informal lunch meetings with all the staff at the GP office. The agenda at each meeting included discussions on: (i) identification of patients where Holter monitoring can be indicated to perform in GP office; (ii) communication pathway; (iii) management of devices and gathering informed consent; and (iv) additional discussion, including details for summary of activity and payment of fees.

Notes were recorded to identify the main themes that can be important from the GP office and patient perspectives.

By the end of the study, we concluded that the project needed to develop at two levels. 

##### 
A. GP Office Level


This level refers to recruiting new GP offices to undergo cluster randomization, where each GP office would be a cluster, including all doctors and nurses working in the clinic. Clusters would be randomized to either the use of C3+ devices followed by referring patients according to the GP criteria, or to use C3+ devices according to the Cardio-Share model. The evaluation of the value of the Cardio-Share model was then based on: (i) workshops; (ii) questionnaires to GPs; and (iii) semistructured interviews with GPs and nurses, with points (i) and (ii) analyzed according to randomization.

##### B. Patient Level

The following themes were identified as important for the patients and were to be evaluated by questionnaires and further explored by semistructured interviews: How well planned is the evaluation pathway? How confident is the patient? How difficult is it to use C3 devices and what is the level of satisfaction with the complete evaluation? 

##### Qualitative Interviews

A group of six GPs and two nurses, all employed at the same shared clinic in Copenhagen, participated in the feasibility phase of the project as special collaboration partners. There were one GP and one nurse who were interviewed on behalf of them all. Prior to the interviews, the interviewed GPs and nurses had meetings with their colleagues to discuss their experiences with participating in the project and the use of the CS model. The individual interviews were recorded and, afterwards, were transcribed verbatim. The interviews were 40 min long.

## 3. Analysis

The transcribed interviews were coded and categorized inductively by using manifest qualitative content analysis [22,23,24,25], inspired by Graneheim and Lundman [23]. First, two authors (C.L. and D.J.) read the transcribed interviews to obtain a sense of the whole content [19,20]. Second, by using an inductive approach to the data and Nvivo 10 software, the text was sorted into units of meaning that were then condensed and grouped to create categories and subcategories [22]. These are listed in Table 2.

The categories and subcategories were compared and discussed by several authors and an external researcher with expertise (M.A.). Two authors (C.L. and H.D.) reviewed participants’ health records to obtain information about diagnoses and number of hospitalizations.

## 4. Results

A total of 117 patients were referred due to suspicion of arrythmia diagnosis. A total of 63 patients were from the orthogeriatric department, 54 patients consulted the GP with the cardiologist, and 34 patients underwent C3 Holter monitoring. The time from referral to a final C3 Holter report was 30 (13) (mean (SD)) days for patients receiving the *Cardio-Share* model, and 117 (45) days for patients receiving the regular Holter in usual care. Additionally, 63 patients underwent C3 Holter monitoring in the orthogeriatric department. In the same period, 119 patients with suspicion of AF underwent Holter monitoring in a conventional setting in the BFH cardiology ambulatory. 

The patient characteristics are summarized in Table 3. The CHADS–Vasc was not recorded for geriatric patients. Only 30% of the patients from the GPs fulfilled the predefined frailty criteria. Psychiatric diseases emerged as an important cause to hinder the usual evaluation for rhythm disorders, while there was only one case of unintended weight loss as a frailty sign. The frailty criteria are listed in Table 1. 

Additionally, 23% of the *Cardio-Share* GP consultations included cardiology questions other than the management of AF, such as seeking advice for the use of antiplatelets, and advice for the management of patients with complex heart failure, which was no longer managed by the cardiology outpatient clinic.

The GPs reported that the main cause for prompting Holter monitoring was the patients’ symptoms, such as dizziness, palpitations, and a feeling of irregular heartbeats. 

The orthogeriatric specialists reported that the main cause prompting Holter monitoring was associated with perioperative short supraventricular runs and the possible clinical need for a pacemaker. 

In the semistructured interviews, the GPs and nurses from the general practice highlighted the ease of using the C3 compact Holter monitor device. The perception from the GPs was that patients generally had great confidence in the Holter monitor and the collaboration with the cardiologist. In some cases, the Holter monitor was used to reassure younger patients without clinical suspicion of significant arrhythmia. In some cases, it was necessary to repeat the Holter monitoring because the equipment was incorrectly operated.

### 4.1. Overall Experiences from the General Practitioners

Overall, the GPs were positive about using Holter monitoring as a diagnostic tool. “From the beginning, it was a very nice project to be part of. Being able to do a Holter-monitoring is a new function, which has been difficult to access, and it has been cumbersome, and suddenly it has moved quite close” (GP). “It is a feature we have been missing, and therefore it has been very easy to accept the project. So, I only have good things to say. It has been a smooth process. There have been only a few technical challenges” (GP). 

The GPs’ assessment was that the patients benefited from having Holter monitoring in the general practice, instead of going to the hospital. The GPs experienced that the patients had an easier course and that they were happy to avoid the hospital. The collaboration with the cardiologist was good and, therefore, the patients felt safe. 


*“And the patients are very, very confident in what we can offer them. They can be evaluated at home” “The patients say it has been easier. They would much rather come here; they think it’s easier because they are comfortable here. They know where to sit in the waiting room and they have an appointment with their general practitioner”*
(GP).

One of the issues is that patients with suspected AF must appear several times in the hospital in connection with the 24 h monitoring of the heart rhythm. The physically or mentally weakest patients risk dropping out of the evaluation if it becomes too demanding. At the same time, it is this group of patients who are most prone to suffer from AF and, as a result, have an increased risk of stroke. They are noncompliant patients. 


*“And then, there are those who never come to the hospital, no matter how sick they are. We can ask them to come here in the surroundings they know and feel safe. But if they must go to the hospital, they will never show up. There are also citizens who are cognitively handicapped and are so poorly functioning that they just can’t; they cannot find their way around, they cannot get in there, they do not know how, it is quite simply not an opportunity for them to visit the outpatient’s clinical the hospital”*
(GP).

### 4.2. The General Practitioners and Nurses Experience a High Level of Professionalism from the Cardiologist

The GPs experienced understandable answers from the cardiologist and felt comfortable with the collaboration. “It has worked really well. The cardiologist is very easy to get hold of. She is very, very helpful and it is the nurse, who is associated with the project as well. If they miss my call, I can be sure that they will call me back as soon as possible” (GP).

It is a great advantage for the general practitioners to be able to discuss the patients with the cardiologist. 


*“The collaboration with the cardiologist is safe and satisfactory. Everybody agrees”*
(Nurse).

### 4.3. The Technique Is User Friendly and Easy to Handle for Both Health Practitioners and Patients

Only a few times, the GPs experienced technical problems. However, this was solved by giving the patient a new period with the Holter monitor.


*“I think we’ve got a single or two files where the recording has not been there. The Holter has been disconnected. Typically, they wear it for more than a day, so you are able to see some of the files, nevertheless”*
(Nurse).


*“I think there have been a few individual episodes where the Holter has been malfunctioning. But then the patient is wired again. So, there are very few technical problems”*
(Nurse).

### 4.4. Benefits of the C3 Holter Monitoring 

The GPs experienced great satisfaction in the collaboration with the cardiologist and with being part of the Cardio-Share feasibility study. The GPs experienced improved work efficiency through communications with cardiologists, thus eliminating work hours.


*“Normally, the patient should be referred to the hospital, when the patient should have Holter-monitoring. We have had direct access to a Holter. Also, there is a quick communication route to get permission. You save an incredible amount of health professionals’ time. In reality, I think it’s a huge financial saving in reality”*
(GP).

In general, the GPs and nurses were very satisfied with being part of the study. It has provided a professional gain. In addition, it made sense for the GPs that the Holter monitoring takes place in the primary sector. 


*“I think Holter-monitoring belongs excellently out here in the primary sector, as long as we are able to get professional supervision with a cardiologist”*
(GP).

## 5. Discussion

### 5.1. Main Findings

A total of 54 patients were included in the feasibility phase of the project. Each case was evaluated in terms of the workflow throughout the diagnostic process, and the patient meeting the criteria of being “frail elderly”. It became clear very fast that being diagnosed using the Cardio-Share model is an advantage not only for the frail elderly, but also for other patients, including younger busy people, or frail patients who are less than 65 years of age. Consequently, not only the frail elderly were included in the main study.

Holter monitoring with a simple compact device is feasible in GP offices, with remote, quick, and efficient support from the hospital-based cardiologists. The healthcare professionals at the general practice were satisfied with the new cross sector, “*The Cardio-Share*” model. A positive picture emerges from the analysis of the interviews with the GPs. The ease of having a C3 compact Holter device is appreciated because it is cumbersome to access Holter monitoring in the hospital ambulatory (BFH). The perception from the GPs is that patients generally have great confidence in the project and the collaboration with the cardiologist. In some cases, the Holter was used to reassure younger patients without suspicion of significant arrhythmia. In some cases, there was a need to reperform the Holter monitoring because the equipment was not correctly operated, which could be avoided in the future by a more thorough introduction and training of the GP staff.

### 5.2. Interpretation of Findings and Relation to Other Studies

Our results find that the time from referral to a final Holter report was 7 (6) (mean (SD)) days using the *Cardio-Share* model, and 49 (34) days in usual settings.

Our results are consistent with the number of patients included in studies for better diagnosis among the frail elderly [26,27]. The GPs experienced that the patients had an easier course and that they were happy to avoid the hospital. The collaboration with the cardiologist was good and, therefore, the patients felt safe. In a qualitative study of patient experiences with newly diagnosed atrial fibrillation at the hospital, Thrysoee et al. found that the patients visiting the hospital were overwhelmed, the information was difficult to understand, and the patients found it difficult to be involved in the decision making [28].

### 5.3. Strengths and Limitations

Only one-third of the patients included in this project actually fulfilled the inclusion criteria. Nevertheless, our focus was the evaluation of the “Cardio-share” model. It was therefore acceptable for the cardiologist to approve patients for evaluation who did not fulfill the frailty criteria. 

The fact that the two staff members interviewed spoke to the entire team beforehand and discussed their experiences regarding the new care model probably introduced significant bias. Therefore, it was considered very important to evaluate the acceptability of the Cardio-Share model by making equally available the C3 devices to upcoming GP clinics and randomizing them by clinic as one cluster: to use the Cardio-Share model (intervention group) or to decide further patient management with cardiologist advice (control group).

### 5.4. Implications for Practice and Future Research

Data on the assessment time and characterization of the patient population during the feasibility phase were used to calculate the number of patients to be included in the subsequent cluster-randomized study. In such a future study, participants should be randomized to the “Cardio-share” model or to the standard assessment procedure for AF in the cardiology ambulatory at the hospital.

## 6. Conclusions

The Cardio-Share model, which utilizes the compact C3 devices for Holter monitoring, is both feasible and acceptable to the health professionals for diagnosing heart-rhythm disorders in an orthogeriatric department and in the GP office in collaboration with a hospital ambulatory. The elapsed time from referral to the final Holter report performed for diagnosis was significantly reduced. We conclude that the health professionals accepted the Cardio-Share model. Additionally, the time used for diagnosing or rejecting possible arrhythmia was reduced.

## Figures and Tables

**Figure 1 ijerph-19-07383-f001:**
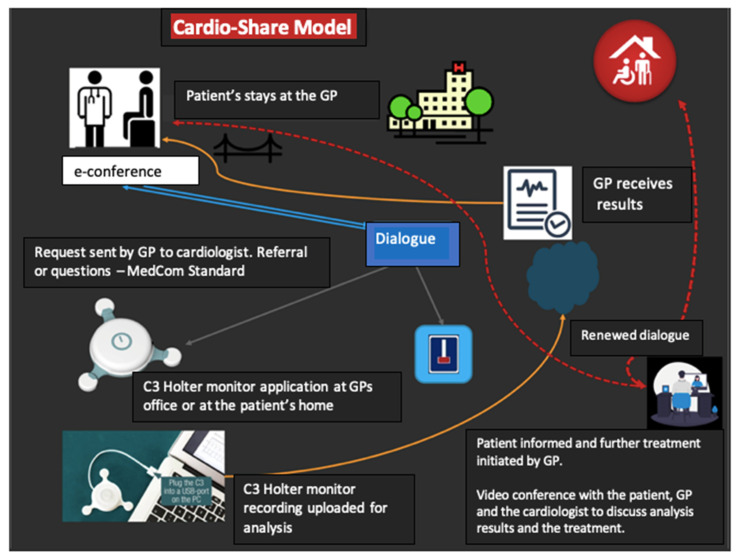
Cardio-Share model. “Dialogue” is exchange of messages requesting further information or clinical data. Yellow lines indicate Holter clinical data flow. Red lines indicate direct virtual communication between the cardiologists, general practitioners, and patients and their caregivers.

**Table 1 ijerph-19-07383-t001:** Frailty criteria.

1	Need for help with transportation to the hospital clinic
2	Need for help with personal hygiene
3	Walking impairment (reduced ability to walk—estimated to take more than 5 s for the patient to walk 5 m)
4	Unintentional weight loss within the past year
5	Cognitive difficulties (dementia, memory problems, aphasia, etc.)
6	Social problems due to alcohol abuse or other abuse, ethnic background, language, etc.

**Table 2 ijerph-19-07383-t002:** Categories and subcategories of GPs’ experiences of the Cardio-Share model.

Categories	Subcategories
Overall experiences of the general practitioners	Collaboration between general practitioner and cardiologist
	Implantation of the C3
	Staff training
	Cardio-Share model
The general practitioners experience a high level of professionalism from the cardiologist	Equipment
	Professionalism
	Confidence in the recordings
	Preferable to use instead of the hospital
The technique is user friendly and easy to handle for both health practitioners and patients	Use of the equipment (C3)
	Use of software
	Patient guidance Technical errors
	Handling problems with C3
	Use of the equipment Introduction and guidance Preparation software Upload software
Benefits of the C3 Holter monitoring	Thoughts on quality
	Opportunities for improvement Benefits of working with a cardiologist

**Table 3 ijerph-19-07383-t003:** Patient characteristics for patients who underwent C3 Holter monitoring.

	GP *	Geriatrics
N	34	63
Age (years, range)	73 (65–90)	83 (55–98)
Proportion age > 75 years (%)	47	77
Gender (% females)	59	53
Number of frailty criteria (mean, range)	2 (1–4)	3 (2–5)
Proportion of psychiatric frailty component (%)	33.3%	35.3%
CHADS–Vasc (mean, range)	3 (1–5)	
Heart failure (N, %)	22 (65)	
Hypertension (N, %)	22 (65)	
Age 65–74 years (N, %)	15 (44)	
Age > 74 years (N, %)	19 (56)	
Stroke (N, %)	3 (9)	
Vascular disease (N, %)	6 (18)	

* Includes only patients of ages > 64 years of a total of 54 patients.

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
