# Peer review of "Reaching Frail Elderly Patients to Optimize Diagnosis and Management of Atrial Fibrillation (REAFEL): A Feasibility Study of a Cross-Sectoral Shared-Care Model"

_ijerph, 2022, doi:10.3390/ijerph19127383_

Round 1

Reviewer 1 Report

Dear Authors, 

Thank you for the opportunity to review the manuscript. Although the care model is not an entirely new concept, there are merits to publish this study as they help us generate greater understanding about the use of technology to improve the AF care. I like the model and description of the technologies used by the authors. This information has been helpful for other clinicians to consider adopting for their practice.

Quality of presentation:  Overall, this study has been presented appropriately. Structure was organized, logical and easy to read. There are some grammatical errors and use of point forms which could affect the readability and quality of the manuscript. I would recommend the authors to seek proof-reading to enhance this.

Ethic approval – is there any ethic approval or waiver?

I have included several specific suggestions for the authors to consider:

L21 – Why is the term “Primary Care” capitalized?

L27 – Revised “JAN-2019 to FEB-2020”.

L46 – There is no need to abbreviate OAC if it is only used once.

L47 – AC is abbreviated. Need to spell in full for people who might not be familiar with the term.

L53, put a common after the “(Figure 1)”.

Figure 1 – “e-konference” is not an English word. Can you revised it in the figure?

Figure 1 – There is an icon below the “Dialogue” box. I am not sure what that icon means. The lines are also in different colors and pattern which I am unsure of its meaning. May I suggest for the authors to consider putting in a legend box to define them for the readers.

Section 3: Method – The stated aim “The aim is to identify patients, who can profit from a faster AF diagnosis with reduced burden related to multiple visits to the cardiology clinic” does not corelate to the study outcomes, especially the qualitative results. May I suggest to the authors to revise the study aims.   

L95 – 96 – revised the phrasing of dates.

Abstract; L74 – 75; L93-95; The authors mentioned in the “Abstract” that the study was a mixed methods design following exploratory sequential scheme…  for a subsequent pragmatic clinical trial”. This could be very confusing for readers. There is a need to define clearly which component comes first, e.g. quantitative ->> quali or quali ->> quant

The “quantitative” aspect of the study does not seem very obvious to me, except for the descriptive statistics used and “time from referral to final Holter report” part. The analysis (L131-136) seems to only describe how you analyze the qualitative data. How about the “quantitative data”?

L78 – 91 – Please avoid writing in point forms. Is this the authors’ study protocol? If so, this section might require some rephrasing.

L115 – 117 – this section would benefit from greater description of your data collection procedures for the qualitative aspect.

Table 1 – The information is not aligned correctly.

Section 4 – analysis. How many GPs and nurse were interviewed here? How many interviews? Individual or group? How many interviews were recorded? How long were the interview? How does the author assure of the “trustworthiness” of their analysis? These information is crucial for qualitative study.

L146 – 148 – “Additionally, 63 patients underwent C3 Holter monitoring. In the same period, 119 patients with suspicion of AF underwent Holter monitoring in a conventional setting in BFH cardiology ambulatory”. Where is this 63 patients from? The geriatric setting?

Table 2 – “proportion age” is the “30” and “70” in the form of percentage or absolute count?

Section 5 – Results. I have a few comments/ suggestions to enhance the qualitative results:

  1. All quotations should be identified to the specific participant – e.g. “the collaboration with the cardiologists…” (GP 1).
  2. The 4 main findings were mostly positive. However, the authors did not mention the limitations. This could raise question about the “over-optimism” and “bias” about their care model without presenting any limitation.

Discussion – I understand that there are similar care models by other researchers. It will be good for the authors to describe and compare their care model with those of other studies too. Example:  David Stevens, Stephanie L Harrison, Ruwanthi Kolamunnage-Dona, Gregory Y H Lip, Deirdre A Lane, The Atrial Fibrillation Better Care pathway for managing atrial fibrillation: a review, EP Europace, Volume 23, Issue 10, October 2021, Pages 1511–1527, https://doi.org/10.1093/europace/euab092

Author Response

Dear Reviewer #1. 

Thank you so much for your helpfullness. We are very pleased to read your comments.  

I have updated the article with your comments and suggestions. 

L21 – Why is the term “Primary Care” capitalized? - We change it to primary care. 

L27 – Revised “JAN-2019 to FEB-2020”. - Accepted

L46 – There is no need to abbreviate OAC if it is only used once. - Accepted

L47 – AC is abbreviated. Need to spell in full for people who might not be familiar with the term. - Accepted

L53, put a common after the “(Figure 1)”. - Accepted 

Figure 1 – “e-konference” is not an English word. Can you revised it in the figure? - we changed it, thanks for the suggestion.

Figure 1 – There is an icon below the “Dialogue” box. I am not sure what that icon means. The lines are also in different colors and pattern which I am unsure of its meaning. May I suggest for the authors to consider putting in a legend box to define them for the readers. - How about: 

“Dialogue” is exchange of messages requesting further information or clinical data. Yellow lines indicate Holter clinical data flow. Red lines indicate direct virtual communication between the cardiologists, the General Practitioners and the patients and their caregivers. 

Section 3: Method – The stated aim “The aim is to identify patients, who can profit from a faster AF diagnosis with reduced burden related to multiple visits to the cardiology clinic” does not corelate to the study outcomes, especially the qualitative results. May I suggest to the authors to revise the study aims.   - We thank reviewer #1 for this point. We have modified the aim to:  The aim is to explore whether the cardio-share model would be feasible and acceptable for GPs to provide  patients  a faster AF diagnosis with reduced burden related to multiple visits to the cardiology clinic. 

L95 – 96 – revised the phrasing of dates. - Accepted 

Abstract; L74 – 75; L93-95; The authors mentioned in the “Abstract” that the study was a mixed methods design following exploratory sequential scheme…  for a subsequent pragmatic clinical trial”. This could be very confusing for readers. There is a need to define clearly which component comes first, e.g. quantitative ->> quali or quali ->> quant  - We are a little confused on this comment. Can you explain it? 

L78 – 91 – Please avoid writing in point forms. Is this the authors’ study protocol? If so, this section might require some rephrasing. - Accepted

Table 1 – The information is not aligned correctly. - We changed it. 

Section 4 – analysis. How many GPs and nurse were interviewed here? How many interviews? Individual or group? How many interviews were recorded? How long were the interview? How does the author assure of the “trustworthiness” of their analysis? These information is crucial for qualitative study.   “Additionally, 63 patients underwent C3 Holter monitoring. In the same period, 119 patients with suspicion of AF underwent Holter monitoring in a conventional setting in BFH cardiology ambulatory”. Where is this 63 patients from? The geriatric setting?

Accepted - Instead we wrote: A group of six GP’s and two nurses, all employed at the same shared clinic in Copenhagen, participated in the feasibility phase of the project as special collaboration partners. There were one GP and one nurse who was interviewed. We recorded two separated induvial interviews. The interviews were 40 minutes long. The transcribed interviews were coded and categorized inductively using manifest qualitative content analysis [21-24], inspired by Graneheim and Lundman [22]. First, two authors (C.L. and D.J.) read the transcribed interviews to obtain a sense of the whole content [19,20]. Secondly, using an inductive approach to data and Nvivo 10 software, the text was sorted into units of meaning that were then condensed and grouped to create categories and sub-categories [21].

Table 2 – “proportion age” is the “30” and “70” in the form of percentage or absolute count? - We changed it in the table, so it is clear now. it is in %. 

Section 5 – Results. I have a few comments/ suggestions to enhance the qualitative results:

  1. All quotations should be identified to the specific participant – e.g. “the collaboration with the cardiologists…” (GP 1).
  2. The 4 main findings were mostly positive. However, the authors did not mention the limitations. This could raise question about the “over-optimism” and “bias” about their care model without presenting any limitation. - Accepted

Discussion – I understand that there are similar care models by other researchers. It will be good for the authors to describe and compare their care model with those of other studies too. Example:  David Stevens, Stephanie L Harrison, Ruwanthi Kolamunnage-Dona, Gregory Y H Lip, Deirdre A Lane, The Atrial Fibrillation Better Care pathway for managing atrial fibrillation: a review, EP Europace, Volume 23, Issue 10, October 2021, Pages 1511–1527, https://doi.org/10.1093/europace/euab092

We thank reviewer #1 for this comment and valuable reference, that we have included in the introduction. Since our study addresses diagnose and guidance to GPs rather and we did not explore patient compliance/non-compliance, we regret that cannot provide these data.

Reviewer 2 Report

The authors investigate reaching the frail elderly patients with Cardio-share strategy. This monitoring system is important to detect atrial fibrillation (AF). However, the description of results was quite insufficient to indicate conclusions. Study plan and subject includes a lot of concerns. The reviewer thinks the strongest profit of this study is that Cardio-share management with C3 device can diagnose AF faster than conventional strategy. In addition, authors should compare with the profits of Implantable loop recorder in discussion if possible.

Major

  1. “Suspected AF” in line 120 is unclear. Authors should identify whether include or exclude patients who had already diagnosed AF in inclusion criteria. In addition. It is important to declare whether patients with past stroke were included or not.
  2. Authors mentioned that patients with over 65 years old were included in line 120. However, study participants included under 65 years old in Table 2. Authors must exclude them from analyses.
  3. The meaning of Table 2 was unclear. Authors should remake this table to indicate the characteristics between Cardio-share with C3 devices and without C3 devices (conventional Holter monitoring?). In addition, there was no explanation about time to diagnose AF. The reviewer thought it is the most important result in this study.
  4. As a clinical perspective, authors should list the CHA2DS2-Vasc score and each past history (Age; <65, 65-74, and >75, Sex, Congestive heart failure history, Hypertension history, Stroke/transient ischemic attack/thromboembolism history, Vascular disease history; prior myocardial infarction, peripheral artery disease, or aortic plaque, Diabetes history) as an absolute number and a percentage in Table 2.
  5. Authors must summarize and make Table or Figure from 5.1 to 5.3. Generally speaking, the score system was used this investigation.

Minor

  1. Abbreviation is incorrect. “OAC” means “oral anticoagulant” in line 46 and cannot be shorten in line 47.
  2. How often can C3 devices report AF event? Continuously or once a day? Please mention in introduction.

Author Response

Dear Reviewer #2. Thank you for your comments and suggestions. We have now uploaded an updated word. 

Major 

  1. “Suspected AF” in line 120 is unclear. Authors should identify whether include or exclude patients who had already diagnosed AF in inclusion criteria. In addition. It is important to declare whether patients with past stroke were included or not. 

    We thank reviewer #2 for this point. We have modified the text making more explicit the patient population. History of stroke was not an exclusion critrerium. Nevertheless, evaluation of cardiac source of stroke elicits more that Holter monitoring and was outside the scope of the study, which we also have added in the text.

  2. Authors mentioned that patients with over 65 years old were included in line 120. However, study participants included under 65 years old in Table 2. Authors must exclude them from analyses. â€‹

    We understand and appreciate this point and have modified the table accordingly.

  3. The meaning of Table 2 was unclear. Authors should remake this table to indicate the characteristics between Cardio-share with C3 devices and without C3 devices (conventional Holter monitoring?). In addition, there was no explanation about time to diagnose AF. The reviewer thought it is the most important result in this study. 

    ​We apologize for causing a misundersanding. All patients managed with the Cardio-Share model were examined with C3 devices. Patients examined with conventional Holter attended the heart ambulatory as it is usual care. We have added this explanation in the text. As a clinical perspective, authors should list the CHA2DS2-Vasc score and each past history (Age; <65, 65-74, and >75, Sex, Congestive heart failure history, Hypertension history, Stroke/transient ischemic attack/thromboembolism history, Vascular disease history; prior myocardial infarction, peripheral artery disease, or aortic plaque, Diabetes history) as an absolute number and a percentage in Table 2. â€‹We are thankful to reviewer #2 for this suggestion. We have added the available data in Table 2.
  4. Authors must summarize and make Table or Figure from 5.1 to 5.3. Generally speaking, the score system was used this investigation. 

    We thank Reviewer #2 for the comment. We are not sure what to change here. Can you explain what you mean? 

Minor.

  1. Abbreviation is incorrect. “OAC” means “oral anticoagulant” in line 46 and cannot be shorten in line 47.
    We understand and appreciate this point and have modified it the article.

  2. How often can C3 devices report AF event? Continuously or once a day? Please mention in introduction. 
We thank reviewer #2 for this suggestion. C3 devices record continously. Report is obtained at the end of recording.  

Round 2

Reviewer 2 Report

The reviewer appreciates authors' reply. 
However, the reviewer still has some concerns.

1. What is the significance just to mention about "5.1.-5.4."?
They were just one's opinion. Authors should score like a scale (1-10) each questionary for each GP and nurse, and summarize overall scores compared with conventional method (Holter) as a research article.

2. What is the meaning of Geriatrics group in Table 2?
The reviewer thinks most valuable finding was "Time from referral to a final C3 Holter report was 30 (13) (mean (SD)) days for patients receiving the Cardio-Share model and 117(45) days for patients receiving regular Holter in usual care. Therefore, authors should show the characteristics of regular Holter group (119 patients) with suspicion of AF underwent Holter monitoring in a conventional setting in BFH cardiology ambulatory instead of Geriatrics group.
In addition, the meaning of "63 patients underwent C3 Holter monitoring in the Orthogeriatric department" was unclear in this context. What did they affect the result or conclusion?
Moreover, the number of participants in GP group was different between the text (n=35) and Table2 (n=34). Which is correct?

3. What is the strength of CS with C3?
Authors should compare with insertable cardiac monitor (ICM) in discussion.

Author Response

Dear Reviewer #2,

We are grateful for your guidance and the possibillity to make revisions to our manuscript responding to reviewer #2.

We have now changed the strategy of the manuscript as it describes our feasibillity study as preparation for a subsequent qaulitative study. We have also described the datastream.

Additionally, we have corrected the period of inclusion as it was from Marts 2019 and not January 2019 as we originally wrote by mistake.

Regarding comments and suggestions:

  1. What is the significance just to mention about "5.1.-5.4."?
    They were just one's opinion. Authors should score like a scale (1-10) each questionary for each GP and nurse, and summarize overall scores compared with conventional method (Holter) as a research article.

We appreciate this important comment as we realize that we have not described the process clearly. Section 5.1. is baed on three meetings with the entire staff at the clinic and a final interview. The meetings with the staff were organized as ’’lunch meetings’’ and the staff attenting these was composed of six general practitioners (one under training in family practice). These unformal meetings provided background material for a subsequent questionary and to elaborate semi-structured interviews. The general practitioners who participated in the interview was asked to dicuss the experience from the staff using remote guidance from the cardiologist to decide indecation and results of heart rhythm monitorering with C3+ devices. This is now described in the manuscript.

  1. What is the meaning of Geriatrics group in Table 2?

The reviewer thinks most valuable finding was "Time from referral to a final C3 Holter report was 30 (13) (mean (SD)) days for patients receiving the Cardio-Share model and 117(45) days for patients receiving regular Holter in usual care. Therefore, authors should show the characteristics of regular Holter group (119 patients) with suspicion of AF underwent Holter monitoring in a conventional setting in BFH cardiology ambulatory instead of Geriatrics group.
In addition, the meaning of "63 patients underwent C3 Holter monitoring in the Orthogeriatric department" was unclear in this context. What did they affect the result or conclusion?
Moreover, the number of participants in GP group was different between the text(n=35) and Table2 (n=34). Which is correct?

We understand this important point but at the time we conducted the feasibility study we did not have permission to follow patients in regular settings, therefore we could only get aggregated data on time for evaluation. The reason to choose a cohort from the Geriatric department is that we used the cardio-share model in this cohort which is definitory “frail” where the Geriatric department were handed two C3+ devices and had the possibility to use them following the same remote cardiologist guidance procedure as the GPs. This is mow explained in the manuscript.

The discrepancy on the number of patients is a mistake (typo) and we have corrected it. The correct number is 34 patients.

  1. What is the strength of CS with C3? Authors should compare with insertable cardiac monitor (ICM) in discussion.

We are thankful for this question. C3+ is a compact device with three built-in electrodes that can be connected to regular electrode patches and monitoring starts pressing the single button in the middle of the device. Hence, it is easy to handle without special skills to place the three electrodes in the middle of the chest on patients who undergo monitoring, compared to multiple electrodes that need to be placed correctly, with hanging cables that require special training of the health staff and more comprehensive instruction to the patients. Additionally, it does not require exclusive and costly electrodes. This explanation is now added to the discussion.

Sincerely,

Caroline Ladegaard (+45 28922205, caroline.ladegaard.01@regionh.dk / carolineladegaard@hotmail.com)

On behalf of the corresponding author

Helena Dominguez, MD, PhD, Assoc.Prof.

Cardiology department Y, Bispebjerg and Frederiksberg Hospital, Copenhagen, Denmark

And

Department of Biomedicine, University of Copenhagen, Denmark

+45 22989343

mdom0002@regionh.dk